# Growth and Body Composition in PKU Children—A Three-Year Prospective Study Comparing the Effects of L-Amino Acid to Glycomacropeptide Protein Substitutes

**DOI:** 10.3390/nu13041323

**Published:** 2021-04-16

**Authors:** Anne Daly, Wolfgang Högler, Nicola Crabtree, Nick Shaw, Sharon Evans, Alex Pinto, Richard Jackson, Boyd J. Strauss, Gisela Wilcox, Júlio C. Rocha, Catherine Ashmore, Anita MacDonald

**Affiliations:** 1Birmingham Women’s and Children’s Hospital, NHS Foundation Trust, Birmingham B4 6NH, UK; nicola.crabtree@nhs.net (N.C.); nick.shaw@nhs.net (N.S.); evanss21@me.com (S.E.); alex.pinto@nhs.net (A.P.); catherine.ashmore@nhs.net (C.A.); anita.macdonald@nhs.net (A.M.); 2Department of Paediatrics and Adolescent Medicine, Johannes Kepler University, Kepler University Hospital, Krankenhausstraße 26-30, 4020 Linz, Austria; wolfgang.hoegler@kepleruniklinikum.at; 3Liverpool Clinical Trials Centre, University of Liverpool, Brownlow Hill, Liverpool L69 3GL, UK; r.j.jackson@liverpool.ac.uk; 4School of Medical Sciences, Faculty of Biology, Medicine and Health Sciences, University of Manchester, Manchester M13 9PL, UK; boyd.strauss@manchester.ac.uk (B.J.S.); gisela.wilcox@srft.nhs.uk (G.W.); 5School of Clinical Sciences, Faculty of Medicine, Nursing and Health Sciences, Monash University, Clayton, VIC 3800, Australia; 6The Mark Holland Metabolic Unit, Salford Royal Foundation NHS Trust, Ladywell NW2, Salford, Manchester M6 8HD, UK; 7Nutrition and Metabolism, NOVA Medical School, Faculdade de Ciências Médicas, Universidade Nova de Lisboa, 1169-056 Lisboa, Portugal; rochajc@nms.unl.pt; 8Centre for Health and Technology and Services Research (CINTESIS), 4200-450 Porto, Portugal

**Keywords:** phenylketonuria, body composition, glycomacropeptide, protein substitute

## Abstract

Protein quality and quantity are important factors in determining lean body (muscle) mass (LBM). In phenylketonuria (PKU), protein substitutes provide most of the nitrogen, either as amino acids (AA) or glycomacropeptide with supplementary amino acids (CGMP-AA). Body composition and growth are important indicators of long-term health. In a 3-year prospective study comparing the impact of AA and CGMP-AA on body composition and growth in PKU, 48 children were recruited. *N* = 19 (median age 11.1 years, range 5–15 years) took AA only, *n* = 16 (median age 7.3 years, range 5–15 years) took a combination of CGMP-AA and AA, (CGMP50) and 13 children (median age 9.2 years, range 5–16 years) took CGMP-AA only (CGMP100). A dual energy X-ray absorptiometry (DXA) scan at enrolment and 36 months measured LBM, % body fat (%BF) and fat mass (FM). Height was measured at enrolment, 12, 24 and 36 months. No correlation or statistically significant differences (after adjusting for age, gender, puberty and phenylalanine blood concentrations) were found between the three groups for LBM, %BF, FM and height. The change in height z scores, (AA 0, CGMP50 +0.4 and CGMP100 +0.7) showed a trend that children in the CGMP100 group were taller, had improved LBM with decreased FM and % BF but this was not statistically significant. There appeared to be no advantage of CGMP-AA compared to AA on body composition after 3-years of follow-up. Although statistically significant differences were not reached, a trend towards improved body composition was observed with CGMP-AA when it provided the entire protein substitute requirement.

## 1. Introduction

There are concerns about increasing obesity and consequential long-term comorbidities in both the general and phenylketonuria (PKU) populations [1,2,3]. A reliance on an “artificial” diet may alter normal physiological processes such as the distribution of fat and lean mass, adversely affecting long-term health outcomes [4]. Body composition is a key component of health, and it typically refers to the quantification of body fat and muscle mass—changes that cannot be adequately assessed by body weight or body mass index (BMI) [5,6]. In PKU, reports of body composition are few and there are no long-term prospective studies or systematic/meta-analyses reviewing body composition. Therefore, it is difficult to extrapolate any association between body composition and other factors such as muscular fitness, adiposity and longer-term health outcomes.

In classical PKU, a low-phenylalanine diet requires substantial modification of usual dietary patterns. For most patients with PKU, high-biological-value proteins are excluded (e.g., meat, fish, eggs and dairy products), with low-phenylalanine/phenylalanine-free protein substitutes providing the principle source of obligatory nitrogen, which is essential to maintain metabolic control and enable optimal growth and lean body mass [7,8]. Muscle contributes up to 40–45% of body weight, and skeletal muscle is the largest store of peptides and free amino acids [9]. Reliance on “synthetic” nitrogen sources may compromise body composition; lean mass is dependent on amino acid availability and, compared to natural protein, delivery and utilisation of amino acids from protein substitutes is sub-optimal [10,11,12].

A consistent finding from a systematic meta-analysis in healthy subjects [13] is that lean mass is a strong predictor of bone mass. Lean mass, and therefore bone mass, particularly in teenagers with PKU, may be compromised as it coincides with a time when adherence with diet and protein substitutes is being challenged [14,15]. Additionally, the physiological increase in lean body mass and fat mass differs in adolescent boys and girls, further exacerbating the difficulties in body composition interpretation [16].

The synthetic protein given in PKU is usually derived from amino acids without phenylalanine (AA). More recently, casein glycomacropeptide (CGMP), a bioactive phosphoglycopeptide, has been used as an alternative low-phenylalanine protein substitute (CGMP-AA). It is associated with better palatability, so adherence is improved [17], but it is unknown if this bioactive macropeptide will alter body composition in PKU. The type of protein, its absorption and amino acid composition alters insulin and glucagon responses. Insulin stimulates protein synthesis [18,19], while glucagon increases amino acid catabolism. Some non-PKU studies have shown that nitrogen retention is improved when protein is in the form of oligopeptides compared to whole protein or amino acids [20,21,22]. It is possible that CGMP, a whey-derived macropeptide, may promote nitrogen retention, improving lean body mass synthesis [23] and growth potential in children with PKU.

This 3-year, prospective, longitudinal study in children with PKU aimed to compare the impact of two different sources of protein substitute, AA and low-phenylalanine CGMP-AA, on growth and body composition by comparing height, lean body mass and fat mass.

## 2. Materials and Methods

### 2.1. Methods

Children were included in the study if they were diagnosed with PKU by newborn screening, aged 5–16 years of age, on dietary treatment only and adherent with protein substitute, with 70% of routine blood phenylalanine concentrations within phenylalanine target range for 6 months before study enrolment. Target blood phenylalanine range for children aged 5–12 years was 120 to ≤ 360 µmol/L and for 13 years and older was 120 to < 600 µmol/L, as recommended by the European PKU guidelines [24].

#### CGMP-AA and AA Protein Substitutes

Two types of protein substitute were studied: AA and CGMP-AA. AA were either powders made up with water to a semi-solid consistency or ready-to-drink liquids providing 10, 15 or 20 g of protein equivalent, tailored to a child’s protein requirements. The CGMP-AA powdered protein substitute (a test product via Vitaflo International Ltd. Liverpool, UK) contained 36 mg of phenylalanine for each 20 g protein equivalent and was reconstituted by adding 120 mL of water. Both products had a similar energy profile per 20 g protein equivalent; CGMP-AA, 120 Kcal, 6.5 g carbohydrate and 1.5 g fat; AA, 124 Kcal, 9.4 g carbohydrate and 0.7 g fat. Threonine and leucine were higher in the CGMP-AA product.

### 2.2. Study Design

In this prospective, longitudinal, 3-year study, home visits were conducted 3 monthly collecting dietary information, weight and height. Dual energy X-ray absorptiometry scans (DXA) measured body composition at enrolment and at 36 months. At enrolment, all the children were on AA protein substitute and had a Tanner pubertal assessment. Following the DXA scan, the patients were divided into 3 subgroups:
(1)AA: protein substitute given as AA only;(2)CGMP50: patients tolerating a combination of CGMP-AA and AA;(3)CGMP100: patients tolerating all their protein substitute as CGMP-AA.

Due to the negative impact on blood phenylalanine control, only some children were able to meet their protein requirements using only CGMP-AA [25]. Therefore, in addition to the AA group, a third group (CGMP50) was introduced where a combination of CGMP-AA and AA provided approximately 50% of the protein equivalent intake.

#### 2.2.1. Selection into CGMP-AA or AA Group

The children chose CGMP-AA or AA, depending on their taste preference. Those in the CGMP-AA group entered CGMP50 or CGMP100 groups depending on their phenylalanine blood concentrations.

#### 2.2.2. Dual X-ray Absorptiometry (DXA)

A DXA scan of the total body to assess body composition (fat and lean body mass) was carried out by two trained operators, using a GE Lunar iDXA and Encore ^TM^ software version 13.1 (GE Healthcare, Wisconsin, MD, USA). Trunk thickness and body weight were utilised to ensure that each child was scanned in the most appropriate acquisition mode. Children lay supine on a bed, while the DXA scan was completed. At baseline and 36 months, the following parameters were measured: lean body mass (LBM) g, fat mass (FM) g, % body fat (%BF), weight (kg), height (cm) and body mass index (BMI) (kg/m^2^). Daily quality assurance tests were performed according to the manufacturer’s instructions. The precision of the instrument was calculated as 1.0% for fat and 0.5% for lean in normal-weight subjects.

#### 2.2.3. Anthropometric Measurements

Weight and height were measured by one of two metabolic dietitians. Height was measured using a Harpenden stadiometer (Holtain Ltd., Crymych, UK) and weight on calibrated digital scales (Seca, Medical Measuring Systems and Scales, Birmingham, UK. Model 875); weight was measured to the nearest 0.1 g and height to the nearest 0.1 cm. Weight, height and BMI were analysed over four time points; baseline, 12, 24 and 36 months.

#### 2.2.4. Blood Phenylalanine Levels

Throughout the study, trained caregivers collected weekly early morning fasted blood spots on filter cards, Perkin Elmer 226 (UK Standard NBS) at home. Blood specimens were sent via first class post to the laboratory at Birmingham Children’s Hospital. All the cards had a standard thickness, and the blood phenylalanine concentrations were calculated on a 3.2 mm punch by MS/MS tandem mass spectrometry.

#### 2.2.5. Pubertal Status

A general medical examination was conducted and pubertal status was measured at enrolment using the Tanner picture index [26]. Stage 1 and 2 were classified as pre-pubertal and stage 3, 4 and 5 as pubertal.

### 2.3. Statistical Analysis of Anthropometry and Body Composition

Continuous data are presented as medians with associated inter-quartile ranges (IQR); categorical data are presented as frequencies of counts with associated percentages. Outcome data were divided into anthropometric data, weight (kg), height (cm) and body mass index (BMI) (kg/m^2^), which were measured as standardised scores, and body composition was measured as lean body mass g, % body fat (%BF) and fat mass g. Anthropometric and body composition data were compared with blood phenylalanine concentrations. Standardised height was represented as the change in height and height z scores at each time point relative to baseline. Given the number of patients and the difference in ages between the groups, analysis was performed using longitudinal regression, which adjusts for patient age. Although standardised measures implicitly account for patient age, it was retained as a covariate in the analysis to avoid any confounding due to age when comparing treatment groups.

Correlations between the outcome data were calculated using Pearson’s correlation coefficient. Anthropometric data were analysed using longitudinal modelling techniques, including main effects for time and age and evaluating the effect of treatment within each time-point. Models were further adjusted for patient age, accounting for the differences in the enrolment age between the treatment groups. Body composition outcomes were measured at two time points; data were analysed using analysis of covariance (ANCOVA) techniques, analysing the 36-month data as the outcome and adjusting for the enrolment data, including patient age at enrolment as well as their pre-pubescent status and gender.

#### Power Calculation

This prospective intervention study took as a primary outcome measure a conservative difference between CGMP and AA groups using day-to-day blood phenylalanine concentrations from a previous study. Twenty children maintaining blood phenylalanine concentrations between 100 and 400 μmol/L would detect a 5% reduction in blood phenylalanine concentrations outside the expected target range, at a power of detection of 88% and at a significance level of *p* = 0.05. A minimum of 45 children was the target recruitment aim.

### 2.4. Ethical Permission

The South Birmingham Research Ethics committee granted a favourable ethical opinion, reference 13/WM/0435 and IRAS (integrated research application system) number 129497. Written informed consent was obtained for all subjects from at least one caregiver with parental responsibility and written assent obtained from the subject if appropriate for their age and level of understanding.

## 3. Results

### 3.1. Subjects

Fifty children (28 boys, 22 girls) with PKU were recruited. Forty-seven children were European and three were of Asian origin. Forty-eight completed the study: 29 in the CGMP-AA group and 19 in the AA group. A significant difference in age was noted between the AA and CGMP50 groups (*p* = 0.005) and between the CGMP50 and CGMP100 groups (*p* = 0.04) (Table 1).

### 3.2. Pubertal Status 

Pre-pubertal status (stage 1 and 2): 32% (*n* = 6/19) were pre-pubertal in the AA group, 69% (*n* = 11/16) in the CGMP50 group and 62% (*n* = 8/13) in the CGMP100 group.

Late puberty (stage 3 to 5): 68% (*n* = 13/19) in the AA group, 31% (*n* = 5/16) in the CGMP50 group and 38% (*n* = 5/13) in the CGMP100 group.

All had classical PKU, except two with mild PKU based on untreated blood phenylalanine levels at diagnosis and dietary phenylalanine tolerance.

### 3.3. Subject Withdrawal

One boy and one girl (aged 12 and 11 years, respectively) in the CGMP-AA group were excluded from the study, as both were unable to adhere to the study protocol. One failed to return blood phenylalanine samples and both had poor adherence to their phenylalanine-restricted diet.

### 3.4. Protein Substitutes and Phenylalanine Concentrations

We have previously reported the types/manufacturers of protein substitutes taken by the AA, CGMP50 and CGMP100 groups. Similarly, the median phenylalanine concentrations have been reported at baseline and year 3 [27]. Median phenylalanine concentrations were within recommended target reference ranges for children aged ≤ 11 and ≥ 12 years old [24].

The median daily dose of protein equivalent from protein substitute was 60 g/day (range 40–80 g), and the median amount of prescribed natural protein was 5.5 g protein/day (range 3–30 g) or 275 mg/day of phenylalanine (range 150–1500 mg) in all groups.

### 3.5. Body Composition Lean Mass, Fat Mass and % Body Fat 

Body composition was analysed using ANCOVA, adjusting for patient age, gender, phenylalanine concentration and pre-pubescent status (Table 2). No statistically significant differences were found between the three treatment groups for lean body mass, %BF or fat mass. All parameters increased over the 3-year study period.

### 3.6. Lean Body Mass, % Body Fat and Fat Mass

ANCOVA showed no significant differences in lean body mass, fat mass or % body fat between the treatment groups, although a trend for improved lean body mass, fat mass and % body fat was observed in the CGMP100 group.

### 3.7. Changes in Height Z Scores 

Accounting for the age and gender differences, there were no statistically significant differences for height within or between the groups. At the end of the 3-year study, all groups had a positive height z score. We have previously reported weight and BMI z scores over the 3 year period [27], showing no statistical differences between the groups (Table 3 and Table 4, Figure 1).

Statistical modelling showed a trend in the CGMP100 group towards improved growth and a reduction in total body fat percentage and improved lean body mass. Analysis of the delta change in height was divided by age for those ≤ 10 and ≥ 11 years. 

## 4. Discussion

Body composition and height over 36 months in a group of children with PKU taking CGMP or AA protein substitutes showed no statistically significant changes in any of the measured parameters. However, in the CGMP100 group, statistical modelling indicated a trend (*p* = 0.42) towards improved longitudinal growth, a reduction in fat mass and % body fat and improved lean body mass. When growth was represented as a median change from baseline over time, it showed that the CGMP100 group had the greatest change in height. However, age modified this trend and, although the CGMP100 group continued to show improved height growth, it did not reach significance.

We can only speculate about a suggested trend in improved body composition when taking CGMP as the complete source of protein substitute. One possible explanation is the bioactivity of CGMP; it is rich in the branched chain amino acids isoleucine and leucine, which are potent modulators of protein turnover and have been shown to have a significant effect on insulin and glucose metabolism [28,29]. If CGMP, by the action of these amino acids, improves insulin sensitivity, it is possible that growth may be improved. However, we did not collect any information on insulin resistance in this group of subjects, so it is difficult to draw any firm conclusions.

Huemer et al. [30] measured growth and body composition over 12 months in 34 children with classical PKU. Total protein intake was 124% of the German recommended daily allowance. A significant correlation was found between lean body mass and intake of natural protein, suggesting that improved natural protein intake was beneficial. Evans et al. [31] also reported a similar significant relationship between a lower % fat mass and a higher total, natural and protein substitute intake, with natural protein > 0.5 g/kg/day associated with an improved body composition; no relationship was found between natural protein intake and improved height z scores. Evans, similarly, to Hoeksma et al. [32], observed that neither natural protein nor energy intake correlated with linear growth, as reported by Aldamiz Echevarria et al. [33]. The effect of a low-protein diet on energy balance and postprandial fat oxidation has received little attention in subjects with PKU. A study by Alfheeaid et al. [34] reported a lower thermal effect of feeding and fat oxidation after healthy subjects had taken a meal containing special low-protein foods and protein substitutes, possibly leading to a higher fat mass and altered body composition. Patients with milder PKU, responsive to sapropterin dihydrochloride (BH4), have a higher natural protein tolerance but there appeared to be no advantage in height, weight, body mass index or growth velocity when BH4 was compared to conventional PKU therapy [35]. There are no studies reporting body composition in BH4-responsive patients who use fewer special low-protein food products and have a wider range of natural protein sources compared to the classically treated patients with PKU.

The importance of adequate protein intake from protein substitutes (both quality and quantity) in patients with PKU has been documented by many authors [36,37,38,39,40,41]. No studies have identified the protein digestibility score or absorption kinetics of CGMP protein substitutes; this is important to ascertain protein efficiency. In healthy adults, protein-containing meals taken at regular intervals improve skeletal muscle protein by 25%, reinforcing the need to consume protein substitutes in divided doses [42,43,44]. The optimal amount of protein substitute based on free amino acids or CGMP remains undefined, but any factor leading to protein inefficiency may compromise body composition and optimal height and increase the incidence of overweight. Another confounding factor that may affect body composition is the effect of a long-term low-phenylalanine diet higher in carbohydrates, which may be associated with a higher risk of adiposity and insulin resistance [45,46]. All of these factors may lead to under achievement of optimal growth potential in children with PKU [47].

Comparison of body composition by gender, regardless of group, showed that lean body mass was statistically significantly higher in males than females, consistent with reports in the literature (*p* = 0.013) [48]. Fat mass and lean body mass vary with age, gender and pubertal status. Various authors have reported an age-related increase in lean body mass index being more rapid in males compared with females, particularly between the ages of 11 and 16 years, which is in line with the rapid accrual of lean body mass during male puberty [49]. Children gain lean mass disproportionately to height and this is more pronounced in boys compared to girls [48].

A multitude of methods exist for assessing body composition, including DXA, bioelectrical impedance (BIA) and whole-body air displacement plethysmograph (Bodpod), each having their own assumptions, advantages and inadequacies [50]. Unfortunately, there is a lack of standardised reference data, making interpretation and comparison of results challenging. Sensitive and accurate measurements are needed to detect differences in visceral, compared to central, fat accumulation, as ponderal and body mass index alone are unable to detect subtle differences. The gold standard for measuring body composition is a four-compartment (4C) model [51]. DXA has been evaluated against 4C models in children, and although it overestimates body fat by 1–4% depending on age, sex and body size, the correlation compared to a 4C model is good despite the small error [52,53]. Reference data for comparison of body composition parameters are limited; a recent publication by Ofenheimer [54] has produced age and gender specific reference percentiles of body composition parameters for European children and adolescents. Comparison of our data would indicate appropriate body composition for fat and lean body mass when calculated as a median between baseline and 36 months.

There are limitations to this study: we did not have a healthy reference control group or UK-based reference data to compare body composition parameters, an inherent problem when using the DXA for body composition analysis. Endocrine parameters such as bone age or growth hormone were not measured, but they may have explained differences in linear growth. Until kinetic studies are conducted, it is unknown if a peptide compared to amino acids alters the delivery and assimilation of amino acids, leading to improved lean body mass and growth. We did not collect parental height data, which may have been useful as a comparison within the groups. Not all the children were able to completely replace their full amino acid requirement with CGMP-AA, which may have reduced the strength of our findings. There were small numbers of children in each study group. More older children chose to stay on their AA supplement compared to the younger age group, who were more willing to try an alternative protein substitute, which may have led to some bias.

## 5. Conclusions

In this 3-year longitudinal study, we found no noticeable differences in body composition between the groups taking CGMP-AA and AA. However, there was a trend towards improved body composition in the group taking all of their protein substitute as CGMP-AA. This may suggest that CGMP does confer some biological benefit. Proof of concept will only be possible via larger controlled studies and over a longer duration throughout childhood.

## Figures and Tables

**Figure 1 nutrients-13-01323-f001:**
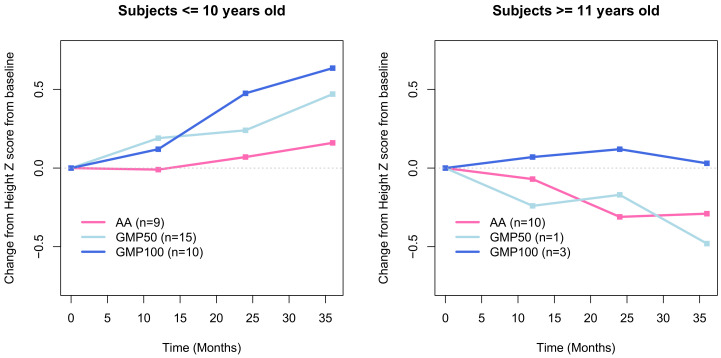
Change in median height z score from baseline to 36 months in the amino acids (AA), GCMP100 and CGMP 50 groups. Legend: AA, amino acid; CGMP, glycomacropeptide; CGMP50, protein substitutes based on combined CGMP and AA; CGMP100, protein substitute based on CGMP only.

**Table 1 nutrients-13-01323-t001:** Subject characteristics at recruitment.

	AA	CGMP50	CGMP100
Number recruited	*n* = 19	*n* = 16	*n* = 13
Girls	*n* = 8	*n* = 8	*n* = 5
Boys	*n* = 11	*n* = 8	*n* = 8
Median age y (range)	11.1 (5–15)	7.3 (5–15)	9.2 (5–16)
% of children prepubertal (stage 1 and 2)	32%	69%	62%
Girls	*n* = 2	*n* = 6	*n* = 5
Boys	*n* = 4	*n* = 5	*n* = 3
% of children pubertal (stage 3 to 5)	68%	31%	38%
Girls	*n* = 6	*n* = 2	*n* = 0
Boys	*n* = 7	*n* = 3	*n* = 5

AA, amino acid; CGMP, glycomacropeptide; CGMP50, protein substitute based on combination of CGMP and AA; CGMP100, protein substitute based on CGMP only.

**Table 2 nutrients-13-01323-t002:** Median (range) lean mass, fat mass and % body fat in the AA, CGMP50 and CGMP 100 groups at enrolment and 36 months.

Body Composition	Time of Assessment	AA (Range) *n* = 19	GMP50 (Range) *n* = 13	GMP100 (Range) *n* = 16
Lean mass (g)	Enrolment	26,702 (16,920–34,209)	16,334 (14,280–17,686)	20,060 (16,451–21,947)
	36 m	32,560 (25,893–40,511)	23,921 (22,725–26,477)	31,268 (25,561–35,875)
Delta		5858 (8973–6302)	7587 (8445–8791)	11,208 (9110–13,928)
Fat mass (g)	Enrolment	9528 (6961–15,018)	5764 (4504–6758)	6688 (5057–8811)
	36 m	17,216 (10,930–20,687)	12,945 (10,678–16,519)	12,220 (8347–13,101)
Delta		7688 (3969–5669)	7181 (6174–9761)	5532 (3290–4290)
% body fat	Enrolment	29 (23–36)	24 (22–28)	25 (19–30)
	36 m	35 (25–39)	33 (30–36)	28 (20–33)
Delta		6	9	3

AA, amino acid; CGMP, glycomacropeptide; CGMP50, protein substitute based on combination of CGMP and AA; CGMP100, protein substitute based on CGMP only; g, grams; kg, kilograms.

**Table 3 nutrients-13-01323-t003:** Median z scores (range) for height in AA, CGMP50 and CGMP100 groups measured annually from enrolment to 36 months in PKU children.

Time (Months)	AA Height z Score *n* = 19	CGMP50 Height z Score *n* = 16	CGMP100 Height z Score *n* = 13
Enrolment (range)	0.2 (−0.2 to 0.8)	−0.1 (−0.6 to 0.6)	−0.1 (−0.4 to 0.3)
12 months (range)	0.2 (−0.2 to 0.6)	0.1 (−0.4 to 0.5)	0.1 (−0.1 to 0.3)
24 months (range)	0.2 (−0.1 to 0.5)	0.2 (−0.2 to 0.5)	0.4 (0.0 to 0.7)
36 months (range)	0.2 (0.0 to 0.5)	0.3 (−0.1 to 0.7)	0.6 (0.1 to 0.7)
Delta height z score	0	+0.4	+0.7

AA, amino acid; CGMP, glycomacropeptide; CGMP50, protein substitute based on combined CGMP and AA; CGMP100, protein substitute based on CGMP only.

**Table 4 nutrients-13-01323-t004:** ANOVA regression model showing values for age, treatment (AA, CGMP50, CGMP100) and treatment/time.

	Df	Sum Sq	Mean Sq	*F* Value	Pr (>F)
Age	1	0.015	0.015	0.020	0.887
Treatment	2	1.225	0.613	0.843	0.432
Treatment/time	3	2.081	0.694	0.954	0.415
Residuals	185	134.418	0.727		

1 = AA, 2 = CGMP50, 3 = CGMP100.

## Data Availability

Data available at the request of the corresponding author.

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
