# Peer review of "Growth and Body Composition in PKU Children—A Three-Year Prospective Study Comparing the Effects of L-Amino Acid to Glycomacropeptide Protein Substitutes"

_nutrients, 2021, doi:10.3390/nu13041323_

Round 1
Reviewer 1 Report
Many of the point addressed could not be satisfied by the authors, as they did not have the required info.
However, the overall merit of the manuscript is evident and publication in the present form can be suggested.
Reviewer 2 Report
I thank authors for answering all questions. I would suggest to make available to most interested readers tables reporting analytical content of tested formulations. This paper is worth to be published.
This manuscript is a resubmission of an earlier submission. The following is a list of the peer review reports and author responses from that submission.
Round 1
Reviewer 1 Report
There are several main issues that make the paper unacceptable for publication. Data reported are too weak and potentially invalidate by several factors:
- Information about the disease severity are missing, have the children phenotype classified as mild, severe or intermediate? These data are crucial to correctly interpret the results from the study and might represent important confounding factors on the outcome,
- do the author are aware about the genetic background of the children? This might be an additional confounding factor
- the results obtained (although not significant!) might be related to nutritional plan administrated to the children and their adherence to the diet rather than to the supplementation. Do the authors have info about the diet?
- the number of patients in the three groups is relatively small
Author Response
Reviewer 1
There are several main issues that make the paper unacceptable for publication. Data reported are too weak and potentially invalidate by several factors:
Point 1- Information about the disease severity are missing, have the children phenotype classified as mild, severe or intermediate? These data are crucial to correctly interpret the results from the study and might represent important confounding factors on the outcome,
We agree that the phenotype will contribute to how the data is interpreted. Unfortunately, we do not have specific information on disease severity such as the genetic mutations. However, this should not exclude this data from publication. There are several published papers without this specific information, and it does not make the data any less reliable or robust.
None of these children were treated with the drug Sapropterin. They were all diagnosed by newborn screening. The newborn screening blood phenylalanine levels fall within the defined guidelines for classical PKU with the exception of two children. We have stated this fact in the manuscript. This is also reflected in the natural protein tolerance. We could have adjusted the manuscript to include this relevant data, showing at diagnosis phenylalanine concentrations were justifiably showing a severe PKU phenotype. Additionally, blood phenylalanine concentrations were within the European PKU guidelines target ranges, (regardless of phenotype) treatment was adequate for all the groups.
Point 2- Do the author are aware about the genetic background of the children?
This might be an additional confounding factor.
As stated above we do not know the genetic background, but we do know that newborn screening levels were (with the exception of two children) meeting the criteria for classical PKU. This is stated in the manuscript.
Point 3- The results obtained (although not significant!) might be related to nutritional plan administrated to the children and their adherence to the diet rather than to the supplementation. Do the authors have info about the diet?
We have previously published work on dietary intake in this group of children (Daly 2020). We have not included this information in the manuscript but can make reference to this data. There were no differences in energy, carbohydrate, fat and fibre intakes in the groups. Similarly, protein substitute provided a median of 85% of the total protein intake (showing the severity of the PKU population) with no specific differences between the groups. We have also collated data on biochemical blood results over the study period and again there were no differences for any of the measured markers. We can include this information in the paper.
Point 4- The number of patients in the three groups is relatively small
This is in line with all the other previous studies on this topic. PKU is a rare disorder, however, this piece of work is unique following a group of children over 3 years, with the exception of the study by JC Rocha, this is the third largest cohort studied examining body composition.
Table of evidence
Author year |
Number/age of subjects Measurement technique |
Allen 1996 |
n=30 PKU (classical) |
Dobbelaere 2003 |
n=20 PKU (classical) |
Huemer 2007 |
n=34 PKU |
Albersen 2009 |
n=20 PKU (classical) |
Adamczyk 2011 |
n=45 PKU |
Douglas 2013 |
n=59 PKU (classical and mild) |
Rocha 2012 Rocha 2013 |
n=89 PKU |
Doulgeraki 2014 |
n=48 PKU |
Mazzola 2016 |
n=27 PKU n=11early n=16 late |
Sailer 2020 |
n=30 PKU |
Therefore, to reject this work on small numbers seems harsh for a rare disorder.
Reviewer 2 Report
I read "Growth and Body Composition in PKU Children—A 3 Year Prospective Study Comparing the Effects of L-Amino Acid to Glycomacropeptide Protein Substitutes, by Anne Daly , Wolfgang Högler, Nicola Crabtree, Nick Shaw, Sharon Evans, Alex Pinto, Richard Jackson, Boyd Strauss, Gisela Wilcox, Júlio C.Rocha, Catherine Ashmore, Anita MacDonald" with great interest, having dealt with an hypothesis of developing a peculiar AA based formulation for PKU adults in the past. The protocol is technically without flaws, a 3 years long study and 48 children enrolled, means a lot of work and significant numbers. But, I am concerned about CGMP and CGMP-AA efficiency. Thus, I would ask authors to prepare a table describing the exact amino acids composition of both glyco-macropeptide and free AA used in this study. May be they consider those informations obvious, but on site of Glytactin, just as an example, the first ingredient is whey protein isolate, and sodium caseinate and soy proteins appeared after leucine, tyrosine, arginine and histidine, and before tryptophan and methionine. No data about amounts were given. Thus, I could not evaluate AA formulation. I could not understand, also, why casein-glycomacropeptide term was used, certainly I am quite ignorant in PKU treatment field, but casein is not the best protein available as reference for mammals, there is vast literature about why it is not even for mice, and an univocal AA formulation of many different caseins very difficult to establish. Thus, to help clinicians not fully dedicated to this illness, a more detailed list of components of formulations used in this study would help in understanding if failure to find improvements would be due either to insufficient nutritional improvement linked to coupling protein supplementation plus AAs, or eventually to qualitative composition of AA formulation utilized. This is a main point, since I would be interested to evaluate both total amounts of AAs, and also if the said formulation provided, as an example, stoichiometric ratios of EAA not matching at best human needs in those delicate patients. A recent report documented experimentally that supplementation of high doses with arginine is linked to kidney damages in aging animals, and this indicates that AAs formulations suitable for supporting human metabolism should be carefully planned independently by ratios contained in foods. Also, nitrogen intake should be calculated as to provide enough EAAs to match requirements. In children, as in this protocol, intake should also support growth.
Author Response
Reviewer 2
I read "Growth and Body Composition in PKU Children—A 3 Year Prospective Study Comparing the Effects of L-Amino Acid to Glycomacropeptide Protein Substitutes, by Anne Daly , Wolfgang Högler, Nicola Crabtree, Nick Shaw, Sharon Evans, Alex Pinto, Richard Jackson, Boyd Strauss, Gisela Wilcox, Júlio C.Rocha, Catherine Ashmore, Anita MacDonald" with great interest, having dealt with an hypothesis of developing a peculiar AA based formulation for PKU adults in the past.
Point 1-The protocol is technically without flaws, a 3 years long study and 48 children enrolled, means a lot of work and significant numbers. But, I am concerned about CGMP and CGMP-AA efficiency. Thus, I would ask authors to prepare a table describing the exact amino acids composition of both glyco-macropeptide and free AA used in this study.
Thank you for the reviewers’ comments which suggest no major flaws in the work submitted. A table can be submitted and has been shown below with the relevant information requested. This is the amount per 20g protein equivalent, and the median dose would be 60g protein equivalent per day
Table 1. Nutritional composition of CGMP-AA2 and AA protein substitutes.
Protein Substitute |
|
CGMP-AA |
AA |
Nutrients |
Units |
Per 20 g PE Sachet |
Per 20 g PE Pouch |
Calories |
Kcal |
120 |
124 |
Protein equivalent |
g |
20 |
20 |
Total Carbohydrate |
g |
6.5 |
9.4 |
Sugars |
g |
2.2 |
7.8 |
Total Fat |
g |
1.5 |
0.7 |
Docosahexaenoic Acid |
mg |
84 |
134 |
Arachidonic Acid |
mg |
- |
- |
Fiber |
g |
0.1 |
- |
L-amino acids |
|
CGMP-AA |
AA |
|
|
20 g PE |
20 g PE |
L-Alanine |
g |
0.83 |
0.92 |
L-Arginine |
g |
0.96 |
1.5 |
L-Aspartic Acid |
g |
1.31 |
2.37 |
L-Cystine |
g |
0.24 |
0.61 |
L-Glutamine |
g |
2.70 |
- |
Glycine |
g |
1.20 |
2.35 |
L-Histidine |
g |
0.70 |
0.92 |
L-Isoleucine |
g |
1.35 |
1.62 |
L-Leucine |
g |
3.00 |
2.54 |
L-Lysine |
g |
0.80 |
1.67 |
L-Methionine |
g |
0.28 |
0.45 |
L-Phenylalanine |
g |
0.03 |
- |
L-Proline |
g |
1.52 |
1.69 |
L-Serine |
g |
0.96 |
1.04 |
L-Threonine |
g |
2.20 |
1.62 |
L-Tryptophan |
g |
0.40 |
0.5 |
L-Tyrosine |
g |
2.24 |
2.38 |
Taurine |
g |
- |
0.04 |
L-Valine |
g |
1.09 |
1.86 |
CGMP-AA 2: casein glycomacropeptide AA: Phenylalanine-free L-amino acid (PKU Cooler 20,Vitaflo International)
The PKU guidelines review digestibility and bioavailablity of L amino acids in detail and recommend an adjustment of a least 20% more than FAO/WHO/UNU 2007 recommended safe levels of protein intake to compensate for the inefficient utilization of amino acids from protein substitutes.
Point 2- Maybe they consider those information’s obvious, but on site of Glytactin, just as an example, the first ingredient is whey protein isolate, and sodium caseinate and soy proteins appeared after leucine, tyrosine, arginine and histidine, and before tryptophan and methionine. No data about amounts were given.
The amount of exact ingredients can be provided in the product we used the following is provided: casein glycol macro peptide isolate (milk), L-leucine, sucrcose, L-tyrosine, flavourings, thickener, emulsifier (soya lecithin), L-arginine
Point 3- Thus, I could not evaluate AA formulation. I could not understand, also, why casein-glycomacropeptide term was used, certainly I am quite ignorant in PKU treatment field, but casein is not the best protein available as reference for mammals, there is vast literature about why it is not even for mice, and an univocal AA formulation of many different caseins very difficult to establish.
The reason for the use of casein is provided below (although this is not relevant for the paper in question). Casein glycomacropeptide is derived from casein. Casein macropeptide (CMP) is a product of kappa casein (a 64 amino acid macropeptide).
When cheese is produced, the CMP undergoes glycosylation and CMP becomes CGMP casein glycomacropeptide. CGMP is soluble and actually found in the whey effluent and makes up approximately 25% of the whey fraction. It is a bioactive phosphomacropeptide and has many associated bioactive properties.
Point 4- Thus, to help clinicians not fully dedicated to this illness, a more detailed list of components of formulations used in this study would help in understanding if failure to find improvements would be due either to insufficient nutritional improvement linked to coupling protein supplementation plus AAs, or eventually to qualitative composition of AA formulation utilized.
This is an interesting point. All the children were growing well which demonstrates that adequate protein equivalent from the formulas has been provided. Additionally, other work produced by the above team show blood biochemistry is adequate reflecting efficient absorption. The reviewer is correct in making the assumption that as we are providing an artificial form of protein and the efficacy of use may be a factor in preventing normal body composition. However, we have alluded to this point in the manuscript.
This line of argument although interesting is theoretical. The use of AA and CGMP substitutes are essential to PKU management.
Point 5- This is a main point, since I would be interested to evaluate both total amounts of AAs, and also if the said formulation provided, as an example, stoichiometric ratios of EAA not matching at best human needs in those delicate patients.
The amounts of EAAs in the formulas are above that of the FAO/WHO/UNU 2007 recommendations but sustain the body’s needs prevent deficiency. AA formulations have been formulated over many years and the ideal balance is still not elucidated.
This subject is moving away from the main point of this paper, which is on body composition.
Point 6- A recent report documented experimentally that supplementation of high doses with arginine is linked to kidney damages in aging animals, and this indicates that AAs formulations suitable for supporting human metabolism should be carefully planned independently by ratios contained in foods. Also, nitrogen intake should be calculated as to provide enough EAAs to match requirements. In children, as in this protocol, intake should also support growth.
Nitrogen requirement in PKU is provided from the protein substitute formula (AA or CGMP-AA). We agree that they have to be carefully balanced, and the composition carefully formulated. The main topic in this paper is body composition, we have stated in the manuscript the inefficient absorption of L-amino acids. However, to compensate for this factor additional protein substitute is provided.
We appreciate the second reviewers’ comments, suggestions, and knowledge regarding the formulations and their compositions. However, the line of argument is not particularly relevant to the manuscript. We have stated in the paper about the inefficient utilization of these formulations, however, there is a wealth of literature to support their efficacy in this group of children.